ecology

age structure, climate change, grey seal, phenology, population dynamics, sea surface temperature

**Author for correspondence:**
James C. Bull
e-mail: j.c.bull@swansea.ac.uk

# Climate causes shifts in grey seal phenology by modifying age structure

James C. Bull[1], Owen R. Jones[2,3], Luca Börger[1], Novella Franconi[1], Roma Banga[1], Kate Lock[4] and Thomas B. Stringell[5]

[1]Department of Biosciences, Swansea University, Swansea SA2 8PP, UK
[2]Department of Biology, University of Southern Denmark, Odense, Denmark
[3]Interdisciplinary Centre on Population Dynamics (CPOP), University of Southern Denmark, Odense, Denmark
[4]Natural Resources Wales, Martin's Haven, Pembrokeshire, UK
[5]Natural Resources Wales, Bangor, Gwynedd, UK

JCB, 0000-0002-4373-6830

There are numerous examples of phenological shifts that are recognized both as indicators of climate change and drivers of ecosystem change. A pressing challenge is to understand the causal mechanisms by which climate affects phenology. We combined annual population census data and individual longitudinal data (1992–2018) on grey seals, *Halicheorus grypus*, to quantify the relationship between pupping season phenology and sea surface temperature. A temperature increase of 2°C was associated with a pupping season advance of approximately seven days at the population level. However, we found that maternal age, rather than sea temperature, accounted for changes in pupping date by individuals. Warmer years were associated with an older average age of mothers, allowing us to explain phenological observations in terms of a changing population age structure. Finally, we developed a matrix population model to test whether our observations were consistent with changes to the stable age distribution. This could not fully account for observed phenological shift, strongly suggesting transient modification of population age structure, for example owing to immigration. We demonstrate a novel mechanism for phenological shifts under climate change in long-lived, age- or stage-structured species with broad implications for dynamics and resilience, as well as population management.

## 1. Introduction

There are numerous examples of changes to the timing of biological events from wide-ranging taxonomic groups and it is well understood that these shifts can provide sensitive indicators of the effects of climate change, e.g. [1–4]. The causes and consequences of phenological shifts resulting from climate change have become a major area of interest in recent years, across many ecosystems and geographical regions; for examples of recent reviews and syntheses, see [5–9]. Now, a pressing challenge is to develop a robust, mechanistic understanding of how these processes operate [10–12]. This is needed to predict changes in abundance and distribution and to facilitate conservation of endangered or otherwise important species [13].

Additionally, species do not exist in isolation and phenological changes can cascade through biological communities, via trophic, competitive, and mutualistic interactions [14–16]. In particular, mismatches in seasonal events, for example between predator and prey populations or flowering plants and their pollinators, can decouple biological communities and lead to critical transitions in population structure, biological regime shifts, and even collapse of ecosystem services [17–19]. In ecological communities that are strongly regulated by top-down processes, climate change-driven shifts in predator distributions and dynamics are likely to have downstream effects on whole assemblages of species and ecosystem

functions [10,14,20–22]. Therefore, understanding the mechanistic drivers of phenological shifts in key predatory species is of fundamental importance for our ability to meet the global challenge of conserving biodiversity and ecosystem function under climate change.

Here, we focus on grey seals (*Halichoerus grypus*), as a well-studied example of marine mammals, e.g. [23–29]. Like many other seal species, grey seals are iteroparous capital breeders with a high degree of site fidelity [30]. They start breeding at around 5 years old and reproduce annually for potentially several decades, resulting in populations with overlapping generations [31,32]. Marine mammal phenology has been cited as evidence of a major system shift in the Indian Ocean and parts of the Southern Ocean [33] and climate-driven changes in seasonal timing are predicted to have substantial impacts on marine mammal populations themselves [32,34]. There are now observable changes in the timing of seal life history across the Atlantic [35–37], including phenological shifts associated with climate [32], making this an important focal group in their own right, as well as important indicators of the wider effects of climate change.

Phenological shifts in life-history events, including breeding [37–39], pupping [40] and moulting [41] have been reported in many seal species. Many species of seal are known to undergo delayed implantation, or embryonic diapause [42–44], in common with other large carnivore species [45–49]. This has been proposed as the mechanism by which climate acts on pupping phenology in seal species [26]. It is hypothesized that colder sea temperatures invoke a physiological response which delays implantation, thus the mode of action operates at the individual seal level [26]. However, whether temperature alone is sufficient is not certain [49,50], and it is hard to test whether this potential mechanism is responsible for observed population level phenological shifts in highly mobile and elusive species.

An alternative hypothesis has been proposed: that the mechanism of pupping season phenology acts at the population level, through climate-driven modification of the local population age structure, rather than at the individual level, e.g. [26,35]. While the effect of climate change on age structure has been shown to underpin phenological shifts in birds [51,52], and fishes [53], the age structure hypothesis has rarely been tested in mammals, including seals, and not supported where it has [35]. If found to occur in nature, this would open an important new line of enquiry into the mechanisms driving phenological responses to climate change in long-lived predators with age- or stage-structured life histories.

A possible reason why the age structure hypothesis has not been supported in seals previously is a focus on local populations and demographic processes [35]. In closed populations, describing the stable age structure may be enough to understand dynamics, brought about through local demographic processes, such as changes to fecundity and differential survival across age classes. However, along with many age- or stage-structured species, grey seals are known to roam widely [54–56] and even where site fidelity is high, local populations are open to immigration and emigration as a further source of changes to age structure [30]. In grey seals, younger adults have been found to be more likely to remain resident and have smaller home ranges than older adults [57]. This suggests a mechanism where older animals are more likely to move to favourable sites as environmental conditions change, increasing the mean age of the population

at those locations. Coupled with older grey seals having higher weaning success [31], and observations that older individuals tend to reproduce earlier across many taxa, including seals [58–62], it seems reasonable to hypothesize that climate change may act on phenology and population dynamics through modification of age structure, underpinned by movement patterns.

The aims of the current study are to: (i) quantify how changes in the timing and progression of the grey seal pupping season are dependent on climatic drivers throughout a long-term monitoring programme; (ii) identify whether phenology acts at the individual level or population level in a natural setting; and (iii) test the hypothesis that modification of population age structure is the causal mechanism for observed phenological changes. While we do not have data to test whether it is local demography or movement that changes age structure, we focus our modelling on whether stable age structure can account for our findings and discuss the likely role of movement in explaining that.

## 2. Methods

### (a) Study site

The Skomer Marine Conservation Zone (SMCZ) area in Pembrokeshire, south Wales, UK (51°43′55.2″ N, 5°16′33.6″ W) comprises the island of Skomer and mainland Marloes Peninsula (figure 1) and is the location of one of the largest grey seal pupping sites in Wales [54]. Adult female grey seals haul out on sheltered beaches and caves throughout this area to give birth and nurse pups until weaning after about three weeks [55]. The pupping season around Pembrokeshire runs from late August to December (figure 1, inset calendar).

### (b) Seal data

Each year, trained staff carry out surveys most days during the pupping season (typically every 1–3 days from late summer to the end of the year). For this study, records of seal pup counts were used for the period 1992–2018 (27 years) (electronic supplementary material, figure S1). Individual records were kept of each pup's progress from birth to moult (approx. three weeks) following a standardized protocol, e.g. [54]. Details included a unique numerical pup identifier, the location (beach name) and date of first observation, the developmental stage (from 1 to 5) of the pup [56], date of subsequent observations, and a record of neonatal survival until weaning.

On Skomer Island, seal pups were marked with a unique pattern using coloured aerosol sheep fleece marker sprays to help monitor each individual. Additionally, adult seals were identified using their distinctive pelage patterns, which persist as unique identifiers throughout adult life, particularly on females [63]. This approach allowed multi-annual records of individual adult female pupping timing to be compiled. On the Marloes Peninsula, pups are spread across a series of small, inaccessible beaches, surrounded by cliffs. Pup numbers on each beach are low enough for individuals to be recognized by experienced staff on subsequent days without marking. This allows accurate recordings to be made during a given season but follow-up of adults between years was not formalized for Marloes seals [55]. Seal pup analysis for this study was conducted across the whole SMCZ, while analysis of adults in this study was conducted on Skomer seals only. Approximately 72% of the seal pup productivity in this study was on Skomer Island.

Some wider context about SMCZ seals is available as electronic supplementary material, Study species.

Proc. R. Soc. B 288: 20212284

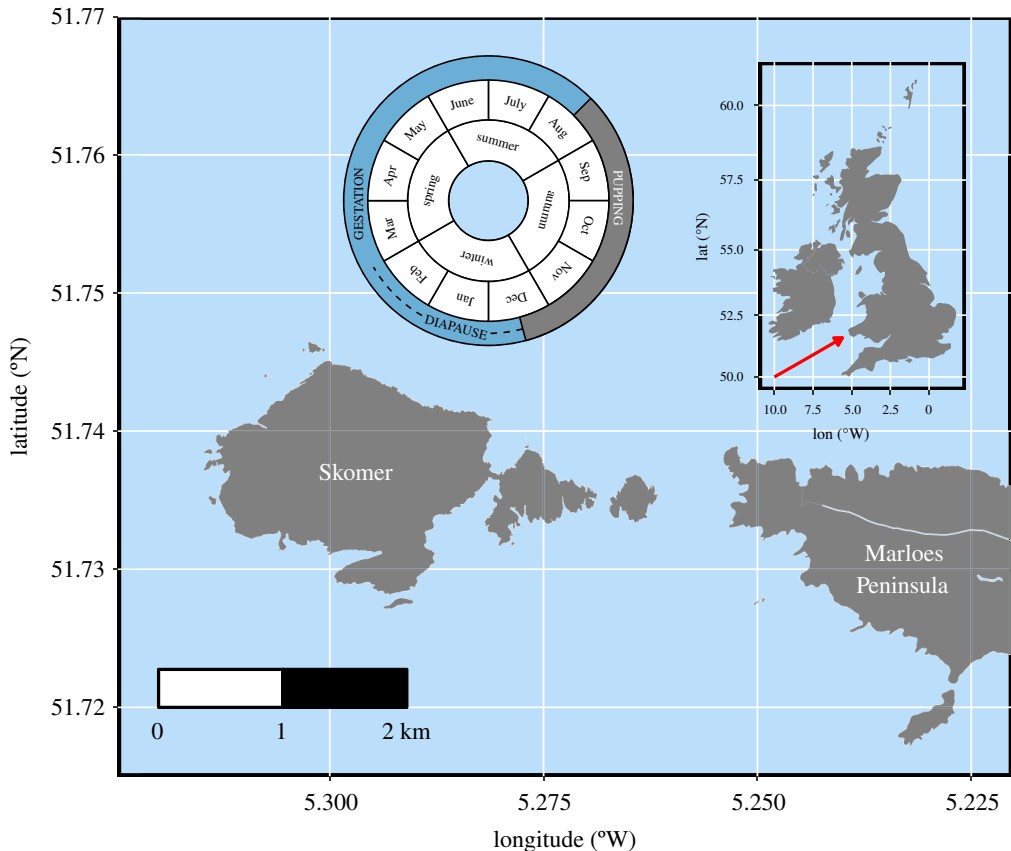

**Figure 1.** Skomer Island and the Marloes Peninsula, southwest Pembrokeshire, Wales. Inset map shows the location of the study area within the British Isles (red arrow). Inset circular plot shows the pupping season for grey seals around Pembrokeshire relative to our defined seasons: 'autumn' = September–November (the majority of the pupping season), 'winter' = December–February, 'spring' = March–May and 'summer' = June–August. (Online version in colour.)

## (c) Environmental data

HadISST1 satellite measurements of sea surface temperatures (SST) over the period of this study were obtained from the British Atmospheric Data Centre (http://www.ceda.ac.uk/data-centres/). These data are resolved into geographical cells (1° latitude × 1° longitude). We used the cell with the northwest corner (52° N, 6° W), covering the whole of the SMCZ. SST data (electronic supplementary material, figure S2) were aggregated into quarterly averages, anchored around the height of the pupping season: 'autumn' = September–November (pupping season), 'winter' = December–February, 'spring' = March–May and 'summer' = June–August.

## (d) Statistical analysis

Observations of seal pupping were used to analyse the cumulative increase in pup counts throughout the pupping season for each year. All dates were described as days since 1 July in a given year. However, because the end of the observing season was dictated more by logistical constraints than the date of the last pup birth, we developed a model to estimate the progression of the pupping season that was insensitive to random fluctuations in the tails of the season.

Empirical cumulative counts of pups were seen to closely follow a sigmoid shape each year (electronic supplementary material, figure S3). Therefore, we fitted three-parameter logistic curves to cumulative pup counts, separately for each year, using nonlinear mixed-effects models with 'year' as a random variable [64]. We accounted for within-year serial dependency as a first-order autoregressive error process, and between-year heteroscedasticity by estimating within-year variances separately. The three parameters that describe these curves are the asymptotic total number of pups, the point of inflection of the fitted sigmoid

curve, and a scale parameter regulating the steepness of increase of the curve. The point of inflection provides a robust estimate of the midpoint of the season (the date where 50% of pups have been counted) and the scale parameter is proportional to the length of the season. We assessed goodness of fit using concordance correlation coefficients [65], with greater than 99.5% concordance in all years.

Changes in each of these three yearly derived parameters (pupping season total, midpoint and length) were modelled separately using generalized additive models (GAMs). We fitted SST as an explanatory variable to test our hypothesis and included the estimate of pupping season total from the previous year as an additional explanatory variable, to account for autocorrelation [37], potentially owing to a number of factors, including autocorrelated environments, energetic carry-overs, or unaccounted for species interactions. Mating occurs after pupping, which could introduce a lagged effect on season midpoint and length. Therefore, we also included the season midpoint and season length estimates from the previous year in our statistical models of season midpoint and length, respectively. All explanatory variables were modelled using cubic smoothing splines. In all cases, we modelled residual error distributions using gamma distributions with natural logarithm link functions.

Here, stepwise model selection was inappropriate, as models are not nested owing to differences in the degree of nonlinearity [66]. In such cases, shrinkage smoothers are recommended [67], which were used here. Separate models were constructed for each of the quarterly SST estimates ('previous autumn', 'winter', 'spring', 'summer' and 'autumn' = current pupping season) and compared using Akaike information criterion (AICc). Since 'winter' was found to result in the lowest AICc, we proceeded using this season. This is also consistent with the hypothesis that the three months following mating (which occurs in the autumn, after pupping) determines the subsequent pupping date [26].

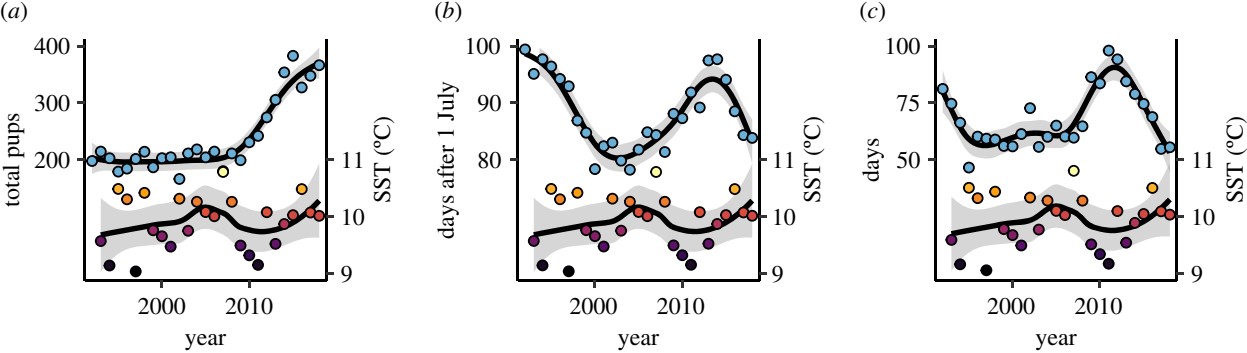

**Figure 2.** Grey seal pupping season parameters and sea surface temperature (SST) around the Skomer Marine Conservation Zone, from 1992 to 2018. Left-hand vertical axes show season parameters: (*a*) the total pups over the season, (*b*) the midpoint of the season (days since 1 July) and (*c*) an estimate of 95% of the season length (days). Blue data points show season parameter estimates. Right-hand vertical scale shows winter SST (December–February). The fill colour of temperature data points (bottom of plots) scales with SST. Solid lines showing fitted trends and shaded ribbons representing marginal 95% confidence intervals. (Online version in colour.)

To assess population-level demographic responses to climate, we used longitudinal data on re-sightings of individual adult females. Here, 'age' is measured as years since first sighting. We modelled how the average age of adult females around Skomer varied with SST using a GAM. We fitted natural logarithm-transformed mean adult age as the response variable, with winter SST as an explanatory variable using cubic smoothing splines. We modelled residual error using Gaussian distributions with identity link functions.

In cases where individual adults were observed to give birth over several years, we were also able to assess whether the date at which an individual gives birth changes with age, or in response to SST, as well as the statistical interaction between ageing and SST. We modelled the date of pupping by individual adults around Skomer using generalized additive mixed models (GAMMs) with gamma error distributions and natural logarithm link functions. We fitted years since the first sighting of a given individual adult and SST as additive fixed effects using cubic smoothing splines. As the age of adults at first sighting is unknown, we also incorporated individual adult female seal identity as a random effect. This allowed us to describe their response to fixed effects without assuming a common starting age (intercept).

### (e) Age-structured population modelling

To explore the relationship between demography and the phenology of the breeding season, we constructed a seven-stage matrix population model (MPM) (electronic supplementary material, table S1). This model serves as a useful tool to explore the potential effects of altered vital rates on population structure, and consequently on population-level breeding phenology. The model is age-based and includes four juvenile stages and three adult stages. The first juvenile stage represents weaned age (approx. 3–4 weeks old) to 1 year, while the others represent years. Reproduction from the three adult stages varies, accounting for the probability of successful weaning depending on maternal age. Therefore, our model's transitions from these adult age-classes to the 'weaning-1 year' stage include both the probability of giving birth and neonatal survival until weaning.

Full details of MPM parameterization are available as the electronic supplementary material, Matrix population model parameterization.

The age structure of the seal population could influence the population-level mean pupping date if individual-level pupping date is associated with maternal age. Therefore, we conducted a sensitivity analysis of this MPM to explore how changes in vital rates, and consequently age structure (stable age distribution), might influence population-level average pupping date. We calculated the expected population-level average pupping date by using

the stable age distribution as weights in a calculation of mean pupping date given our knowledge of age-specific pupping dates.

Specifically, we calculated the change in population-level mean pupping date resulting from a +10% increase in each matrix element. For transitions with a negative sensitivity (i.e. that could result in an earlier pupping date), we then asked how much earlier could the pupping date potentially become? We did this by calculating the effect on pupping date of increasing these survival transitions up to the maximum reasonable value of 0.999.

All statistical analysis was undertaken using R v. 3.6.0 [68] with additional packages: AICcmodavg v. 2.3–1 for model comparisons and selection [69], mgcv v. 1.8–31 for GAM(M)s [70], ncdf4 v. 1.17 for reading SST data from .nc files [71], and popdemo v. 1.3–0 for matrix population modelling [72].

## 3. Results

### (a) Phenological trends

Overall, the SMCZ grey seal pupping season showed marked variation throughout the study period, 1992–2018 (figure 2). The annual total grey seal pup counts showed a substantial increase in the last quarter of the survey period, averaging around 200 pups until 2009, then doubling by 2015 (figure 2*a*). The estimated midpoint of the season was in the first week of October at the beginning of the survey (latest date: 7 October in 1992), advancing by about three weeks to mid-September through the middle part of the survey (earliest date: 16 September in 2004), and returning to early October towards the end of the survey (figure 2*b*). The estimated pupping season length showed a twofold change throughout the monitoring period: 47 days in 1995, 98 days in 2011 (figure 2*c*). For all three phenological parameters, the respective parameter value from the previous year had a statistically significant effect (total pup estimate: $F = 51.7$, $p < 0.001$; season length: $F = 11.4$, $p < 0.001$; season midpoint: $F = 36.1$, $p < 0.001$), demonstrating strong temporal autocorrelation. We went on to test the hypothesis that SST affected pupping season phenology.

### (b) Environmental predictors of phenology

We tested the effects of SST as a predictor of key parameters describing the seal pupping season (table 1). Winter SST was associated with pupping season midpoint: increasing SST was associated with an advance of 7.6 days to the

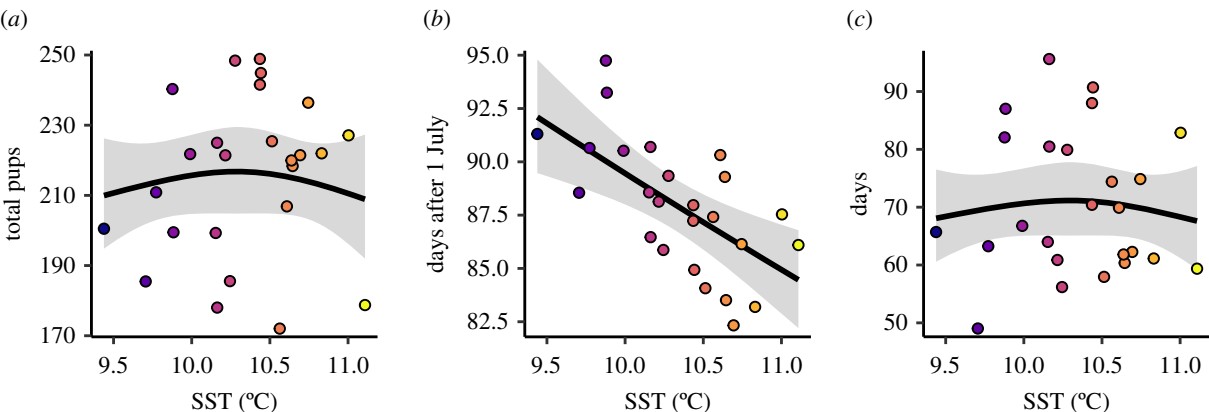

**Figure 3.** Winter (December–February) sea surface temperature (SST) as a predictor of the grey seal pupping season around Skomer Marine Conservation Zone: the effects of winter SST on (a) total pup count, (b) pupping season midpoint and (c) season length. Solid lines show fitted estimates, with shaded ribbons representing marginal 95% confidence intervals, and points are partial residuals. The fill colour of data points scales with SST. (Online version in colour.)

**Table 1.** Winter (December–February) sea surface temperature (SST) as a predictor of the grey seal pupping season around Skomer Marine Conservation Zone. (Season parameters are total pups over the season, the midpoint of the season (days since 1 July) and an estimate of 95% of the season length (days).)

| term | estimates | | | SST | |
|---|---|---|---|---|---|
| | min (year) | max (year) | range | F-ratio | p-value |
| total | 166 (2002) | 383 (2015) | 216 | 0.001 | 0.933 |
| midpoint | 78 (2004) | 99 (1992) | 21 | 2.571 | 0.018 |
| length | 47 (1995) | 98 (2011) | 51 | 0.001 | 0.870 |

season, explaining 36% of the observed phenological shift (figure 3). However, SST did not explain a statistically significant amount of observed variation in pup numbers or season length.

### (c) Individual adult female responses

To understand the phenological patterns that we uncovered, we first explored the prevailing hypothesis; that the mechanism underpinning phenological responses in grey seal pupping operates at the individual adult seal level (through temperature-dependent delayed implantation). Of the adult female seals giving birth to a pup at least once around Skomer, the identities of 150 were recorded (out of a total of 327 adult grey seals identified). Of these, 58 were recorded to have given birth on more than 1 year, and there was no linear trend in the average age of mothers over the monitoring period ($F = 3.79$, $p = 0.084$). In those cases, we assessed the role of adult female ageing (estimated by years since first sighting) and SST on date of pupping over successive years. We found no statistical interaction between ageing and SST ($\Delta AICc = 1.93$), suggesting that individual level phenological responses to SST were not dependent on ageing of the mother. Ageing of the mother had a statistically significant effect (figure 4a; $F = 9.77$, $p = 0.002$), with adults giving birth to pups earlier as they got older, on average, but SST did not affect when an individual gave birth (figure 4b; $F = 0.206$, $p = 0.65$). These findings do not support the prevailing hypothesis that sea temperature drives phenological shifts by act directly on individuals' dates of pupping.

### (d) Population responses

Next, we tested a novel hypothesis that the phenological mechanism operates at the population level, through modification of the local population age structure. Based on our observations that older females give birth earlier (individual adult dataset), and that warmer SST is associated with earlier pupping seasons (pup population dataset), we predicted that warmer SST would be associated with an increase in the average age of females giving birth (individual adult dataset). As predicted, we found average age of adult females increased with increasing SST (figure 4c; $F = 11.8$, $p = 0.002$).

### (e) Age-structured population modelling

Our matrix model provided a very good approximation to dynamics, with a finite population growth rate ($\lambda$) of 1.07; almost identical to the maximum likelihood estimate of growth rate fitted to empirical population size ($\lambda = 1.07$; electronic supplementary material, figure S3, left panel), and very similar to rates reported elsewhere the UK [73] and North America [74]. As expected, the sensitivity analysis showed that an increase in the transition rates for older breeding adult survival (5–6 and 6+ years) led to an earlier mean pupping date (figure 5a). By contrast, an increase in younger breeding adult survival [4,5] led to a later pupping season, demonstrating the timing of the pupping season is dependent on the relative balance of younger and older breeding adults in the population. However, most importantly, we found that manipulating just these two older stages by increasing

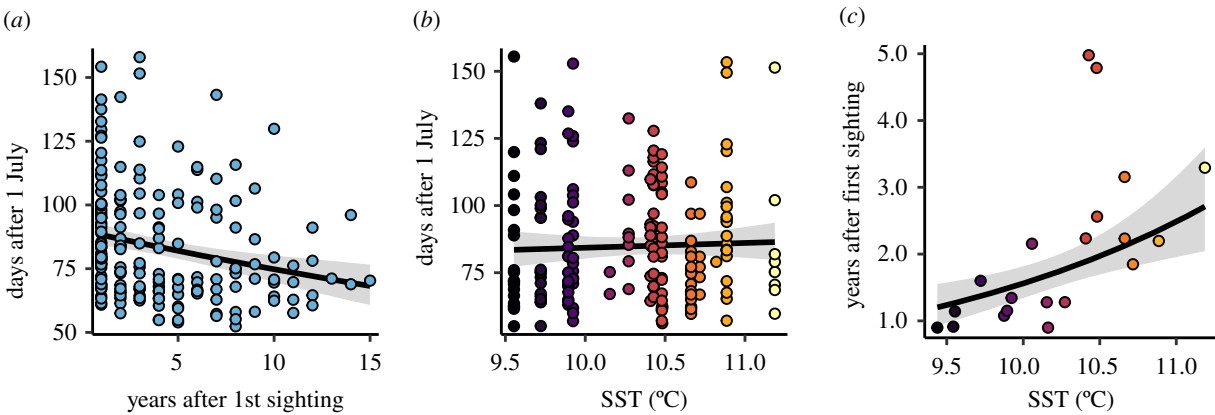

**Figure 4.** Drivers of grey seal pupping timing at the individual level. The effects of (a) adult female ageing and (b) winter sea surface temperature (SST) on the date of pupping by individual females. (c) The effect of SST on the mean age of adult females in the pupping season at Skomer. Solid lines show fitted estimates, with shaded ribbons representing marginal 95% confidence intervals, and points are partial residuals. The fill colour of data points in (b) and (c) scales with SST. (Online version in colour.)

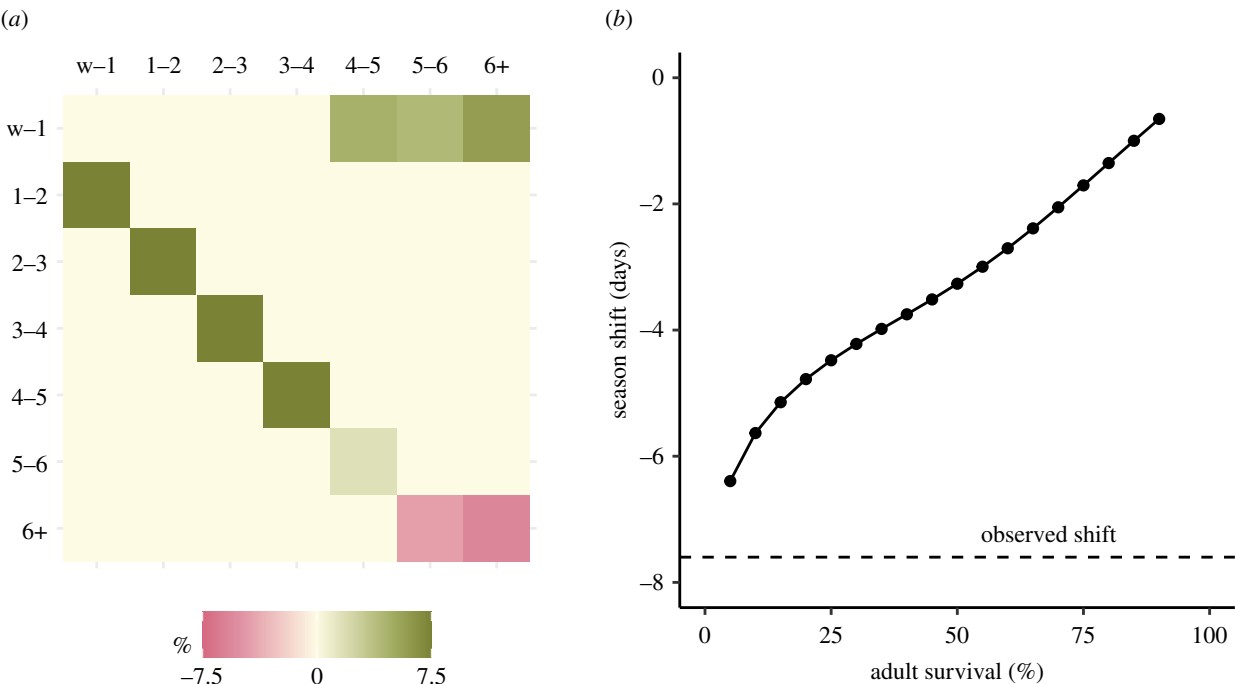

**Figure 5.** (a) The sensitivity of population-mean pupping date. Pink colours represent a negative sensitivity whereby a positive change in the transition rate leads to an earlier pupping date, whereas green colours indicate a later date. The cream colour represents no change (impossible transitions, in this case). There are two transitions (5–6 and 6+ years) where an increase in transition rate (increased survival) result in a stable age distribution that leads to an earlier mean pupping date. (b) Predicted pupping season shift across a range of possible adult survival values for the 5–6 and 6+ transitions. Adult survival was set at 5% increments between 5% and 90%, then increased to 99.9% and the resulting advance in the pupping season compared to the empirical phenological shift (dashed line). (Online version in colour.)

survival to 99.9% could lead to at most a shift in mean pupping date by just −0.65 days (i.e. 15.7 h earlier).

We further investigated the generality of our MPM by testing a range of breeding adult survival values in the two oldest groups (5–6 and 6+ years), from 5% to our original 90%, and calculating the predicted advance to the pupping season when survival was increased to 0.999. For lower adult survival values, the absolute increase to 99.9% is clearly larger, so a greater advance to the pupping season is predicted. However, even when we set adult breeding survival in the oldest two transitions to just 5%, increasing to 99.9% could only produce a shift of −6.4 days, against our observed advance of 7.6 days (figure 5b).

## 4. Discussion

While broad-scale, multispecies studies of ecological patterns are critical to identify global challenges resulting from climate change, a recent review has highlighted a lack of mechanistic understanding in this area [12]. This is needed to help us reconcile observed differences in phenological responses to environmental change across, for example latitude, elevation, trophic level, migratory strategy, thermoregulatory mode and generation time. In particular, where research into phenological shifts moves beyond correlative studies of cue and response, these often assume a single mechanism and do not consider or compare alternatives [12]. This is potentially a

critical shortcoming that now presents a barrier to our ability to mitigate against climate change-induced phenological shifts.

We chose a well-studied and regionally important predator species, the grey seal (*H. grypus*), to test different hypotheses on the underlying mechanism of phenological change. Our key population-level finding was that the peak of the pupping season was more than a week earlier in the warmest years, compared to the coolest. This response was most strongly associated with the previous winter's SST, which we defined as December–February, i.e. SST towards the end of, and shortly after, a given pupping season affects the timing of the following pupping season. This is consistent with the observation that grey seal pupping is earlier in populations that experience warmer SSTs due to latitude [26]. Elsewhere, a negative relationship has also been shown between grey seal birth date and SST, as well as other broad scale climate indicators [32]. The similarity with our population level findings suggests our inferences about the mechanistic basis of this phenological shift may be more widely generalizable.

Our key population-level result is consistent with the mechanism of delayed implantation, set by sea temperatures in the months after mating (which follows autumn pupping in our study population). However, we found no influence of SST in a given year on the timing of pupping for individual mothers. As capital breeders, grey seals accumulate resources for breeding during the majority of the year and then do not forage while suckling their pups [31]. Therefore, the influence of sea conditions earlier in the year is also consistent with resource-limitation affecting competition between individuals. We found that older mothers tended to give birth to pups earlier in the season. This has also been observed in other seal species [58,59], terrestrial mammals [60], as well as many birds [61,62,75].

These observations using population census data provides the basis of an alternative hypothesis based on the age structure of the population: observation (1) warmer years are associated with earlier pupping seasons, and observation (2) older females breed earlier. Therefore, we expected observation (3) the average age of females pupping in warmer years will be greater. We tested this prediction using individual-based, longitudinal data from our focal grey seal population in the SMCZ. Overall, population-level data showed a 1°C increase in observed winter SST equated to around 3–4 days advance in peak pupping date (observation 1). From our individual-level data, we found that a 1-year increase in mean adult female age equated to a 1-day advance in mean pupping date (observation 2). Therefore, we predicted a change of around 6 years to the mean adult female age across the observed temperature range of nearly 2°C. In fact, we observed a difference of just 2 years associated with the observed temperature range (observation 3). However, this discrepancy is within the 95% confidence intervals in parameter estimates so, we believe, represents good support for the age-structured pupping phenology hypothesis.

The SMCZ is a major grey seal pupping site in Wales, so we assume that adult females sighted there are the 'winners' of any competition. Given the time lag with SST, it seems that such competition would be for limiting resources such as prey availability earlier in the year, rather than access to beaches during the pupping season. However, it is known that Welsh seals can travel widely throughout the Irish Sea and as far as southwest England, France and Ireland [30,76–78]. Therefore, a wider study of other pupping sites around the region would be needed to really understand the processes driving dynamics in such a mobile and open population.

Grey seals are particularly well studied in the UK, and similar data to ours have been collected at several Welsh pupping sites, e.g. [79]. It would be possible to infer connectivity between sites using time-lagged spatial cross-correlation of annual pup censuses. However, to fully separate the effects of local demography from movement, direct observation of adult resightings quantifying breeding female site fidelity between years are required. In the Welsh context, the recently published EIRPHOT database of individual adult grey seals, identified from photos, across the Irish Sea would be ideal [30].

The reasons for older females of many species breeding earlier have been explored elsewhere, e.g. [80,81]. This, and associated increases in reproductive success in older females, are hypothesized to result from one or more of: (i) 'selection'—the disappearance of late-breeders over age classes, (ii) 'restraint'—the prioritization of current breeding over future survival in older females, (iii) 'constraint'—improvement in breeding skills with age, and (iv) 'delayed breeding'—the incorporation of early breeders over age classes [82]. Whichever mechanism accounts for an increase in the average age of female grey seals around the SMCZ in warmer years, it is also well documented that older grey seals are typical of long-lived animals, e.g. red deer [83], in having increased success in raising a pup to weaning [31]. Concomitantly, pups born early in the season are more likely to be weaned successfully [84], having potentially important consequences for population resilience and viability with climate change.

We concluded our study by developing an age-structured MPM with which to explore dynamics. This model was motivated by the life history of grey seals and parameterized using grey seal demography data where appropriate [31,85,86]. While our empirical data showed a shift in phenology and age-structure associated with SST, there was no clear, monotonic trend in SST during the study period. Therefore, we question whether SST is acting on the stable age distribution of the population, or through transient effects. If SST affects stable age distribution, then this could affect long-term population growth rate and resilience, since older females are known to have greater weaning success rates [31]. Alternatively, the SMCZ grey seal population is known to mix with neighbouring populations [30], such that SST might change population structure through immigration and emigration. Our MPM clearly showed that biasing the stable age distribution to its limit could not fully account for the observed advance in mean pupping date. This supports the hypothesis that transient changes to the age structure of the population, in response to environmental stochasticity in SST, are currently underpinning observed phenological shifts.

The relationship between age structure, climate change and phenology has been studied across a broad taxonomic range of long-lived animal species, including birds [51,52] and fishes [53], and proposed in mammals [35,87]. The hypothesized underlying mechanism typically focusses on age-dependent differences in local demographic plasticity. For example, in superb fairy wrens (*Malurus cyaneus*), older males are thought to risk moulting in less favourable times of year than younger males [51], in Eurasian blue tits (*Cyanistes caeruleus*) older females are less plastic in their reproductive phenology than younger birds [52], and in walleye pollock (*Gadus chalcogrammus*) climate variation interacts with the age-diversity of the stock to drive mean spawning date [53]. In parallel, climate is known to affect the phenology of long-range movement in

numerous species, which itself may be age-dependent [88–90]. However, our findings indicate that these two research areas, demography and movement ecology, need to be considered together to understand the interplay between population dynamics, climate change and phenology.

In summary, the hypothesis that climate variability acts at the population level, through modification of age structure, provides an alternative to the longstanding, individual-level hypothesis of delayed implantation in large carnivores [35,40]. Moreover, this age structuring mechanism results in population dynamic responses to climate warming that the widely assumed delayed implantation mechanism would not. This may well explain the poor support for a direct link between sea temperatures and pupping phenology in grey seals to date [32,49,50]. More than simply demonstrating a newly recognized phenomenon in a single species, our findings provide motivation and guidance for researchers to consider alternative mechanisms for phenological cue-response shifts in other long-lived species; in particular, including mechanisms acting at the population level.

Ethics. Spray-marking of seals is conducted under published protocols (Alexander M. 2015 Skomer MCZ and Skomer Island Grey Seal management plan; Poole J. 1996 Grey Seal Monitoring Handbook, Skomer Island. Countryside Council for Wales). Contracted surveyors are issued permits by Natural Resources Wales.

Data accessibility. All data used in this manuscript are available as electronic supplementary material [91].

Authors' contributions. J.C.B.: conceptualization, formal analysis, investigation, methodology, project administration, software, supervision, validation, visualization, writing—original draft, writing—review and editing; O.R.J.: conceptualization, formal analysis, investigation, methodology, software, visualization, writing—original draft, writing—review and editing; L.B.: conceptualization, writing—review and editing; N.F.: data curation, validation, writing—review and editing; R.B.: data curation, validation, writing—review and editing; K.L.: data curation, investigation, methodology, writing—review and editing; T.B.S.: conceptualization, funding acquisition, project administration, resources, writing—review and editing. All authors gave final approval for publication and agreed to be held accountable for the work performed therein.

Competing interests. We declare we have no competing interests.

Funding. All work was funded by Natural Resources Wales or its predecessor organization Countryside Council for Wales. All Skomer Island work was completed by the Wildlife Trust under contract by CCW/NRW.

Acknowledgements. We are grateful to many Skomer Marine Conservation Zone, Natural Resources Wales staff and Skomer Island National Nature Reserve, Wildlife Trust South and West Wales staff who have contributed to collecting seal data throughout this long-term monitoring programme.

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
