## [Peer Review File · Proceedings of the Royal Society B: Biological Sciences]

Review History

RSPB-2021-0550.R0 (Original submission)

Review form: Reviewer 1

Recommendation

Major revision is needed (please make suggestions in comments)

Scientific importance: Is the manuscript an original and important contribution to its field?

Acceptable

General interest: Is the paper of sufficient general interest?

Good

Quality of the paper: Is the overall quality of the paper suitable?

Acceptable

Is the length of the paper justified?

Yes

Should the paper be seen by a specialist statistical reviewer?

No

Do you have any concerns about statistical analyses in this paper? If so, please specify them explicitly in your report.

No

It is a condition of publication that authors make their supporting data, code and materials available - either as supplementary material or hosted in an external repository. Please rate, if applicable, the supporting data on the following criteria.

Is it accessible?

Yes

Is it clear?

Yes

Is it adequate?

Yes

Do you have any ethical concerns with this paper?

No

Comments to the Author

The authors try to understand the main drivers behind change in phenology of grey seals. An interesting aspect of the study is coupling the results with a population model to test one of the hypothesis.

However, the study would benefit from additional, broader analysis and further clarification. Grey seals in the UK waters are an interesting subject to study due to its different population dynamics around the UK, with some places still showing almost exponential increase. As also recommended by SCOS, grey seals should be more studied on a broader, geographical range, rather than small isolated populations. It has been shown by many studies that these animals travel a lot between regions, something which authors also mention. This study would, therefore, widely benefit from expanding the currently used dataset to other areas of the UK, where number of seals is much higher. Especially that 'Welsh' grey seals are difficult to study due to their cave breeding behaviour. There is a long term monitoring of individual and population-level breeding rate and phenology in many places around the UK.

It is also difficult to understand the main message of the paper. In some sections the authors focus mainly on changes of SST, which are not very pronounced during the study period (eg Fig S1), in other section on changes in age structure of the population and it is, therefore, not clear how the authors separated the effect of one from another on the phenology.

The study is missing more in depth description (especially for non-seal specialists) how changes in SST (or any environmental change) actually affect age - structure of a population. More information on a potential effect of climate change on fecundity, pup, and juvenile survival would be useful.

Too little effort (in my opinion) is put into explaining how SST or other factors may or may not affect phenology: is it via changes in food availability/distribution/type, is it direct effect on implantation rate, could overall (or local Fig S2) increase in grey seal population affect phenology? Could you maybe use your population model to explore it further? Which life history event of grey seals is most susceptible to changes in SST (e.g. implantation, birth, lactation, survival to age 1) and how it can be translated into population dynamics?

Finally, I find it a little bit concerning that the parameters for the population model are based on a very different seal population than the study population. Are there any data available for the grey seals anywhere around the UK instead of using data from north American seals?

Detailed comments:

Lines 90-92: explain how climate change may affect population age structure

Lines 324-327 These sentences seem to be incomplete. [...] we predicted that warmer SST would

be associated with older females giving birth, on average (individual adult dataset). Does it mean that higher proportion of older females would give birth than during colder years? As predicted, we found average age (of ?) increased with increasing SST ($F = 11.8$, $p = 0.002$, Figure 4). Do younger females have lower fertility or lower weaning success or are just not pregnant at all? I know these are not easy questions to answer

Figure 4. I don't find it clear (based on the figure) that increase in SST results in older females giving birth earlier. Please clarify how do you distinguish between the effect of age and the effect of SST. The slope of the regression on panel a is mainly driven by larger sample size for young females and small sample size for the older females. Although I know that aging adults is difficult only based on visual observation, it is hard to interpret whether a female 1 year after 1st sighting is a female of for example age 4 which would be her first potential breeding season, or an older, more experienced female. No information what colours represent is provided. Your data seem to show quite a large range and variability, yet your CIs are very small.

Line 340. I can't see empirical population size on Figure S3, left panel

Figure 5. You mention three, I can only see two. Not clear what the scale (-0.012 to 0.012) represents. Usually sensitivity is represented as percentage change in relation to final settings of the model.

Line 403. Not really clear what [1] refer to. I guess your 1) in the above paragraph but please clarify

Figure S3. The ribbons are not visible. Are CI so small? It would be great to couple this graph with temporal changes in SST. The midpoint data and the length of the season show a cyclical trend. Could you explain it?

Figure 2: add that y axis show number of pups born per day

Figure 3. What do different colours of points represent? I guess range of temperature but please clarify

A circular graph showing timing of life events would be helpful to show when do pupping, implantation and weaning occur in relation to your quarterly SST. For an example see Fig. 1 in Silva et al (2020), *Environment International* 145

Supplementary material:

Study species section is not referred to in the main text

They show the probability of giving birth is 0.731 for females aged between 4-15 years and 0.831 for those aged between 16-26 years. In our model we assumed used the former figure (0.731) for the first two age classes of breeders, and we used the average of these for the oldest 'open' age group (0.781). How sensitive is your model to your assumptions of the lower age for the 'open' group?

Review form: Reviewer 2

Recommendation

Major revision is needed (please make suggestions in comments)

Scientific importance: Is the manuscript an original and important contribution to its field?

Acceptable

General interest: Is the paper of sufficient general interest?

Acceptable

Quality of the paper: Is the overall quality of the paper suitable?

Good

Is the length of the paper justified?

Yes

Should the paper be seen by a specialist statistical reviewer?

No

Do you have any concerns about statistical analyses in this paper? If so, please specify them explicitly in your report.

Yes

It is a condition of publication that authors make their supporting data, code and materials available - either as supplementary material or hosted in an external repository. Please rate, if applicable, the supporting data on the following criteria.

Is it accessible?

Yes

Is it clear?

Yes

Is it adequate?

Yes

Do you have any ethical concerns with this paper?

No

Comments to the Author

In their manuscript "Climate causes shifts in grey seal phenology through modifying age structure" Bell and coauthors describe an analysis of long term monitoring data on pup production in grey seals in the Skomer Marine Conservation Zone, Wales. The authors find that the timing of pup production shifts with temperature, that older mothers breed earlier, and that warmer years were associated with older mothers. The authors constructed an age structured model to assess whether intrinsic changes in age structure could explain the shift and conclude that the observed changes must be driven by immigration / emigration.

Overall, this manuscript is well-written and most of the analyses seem reasonable. That said, the figures provided feel a bit incomplete. For instance, I would have liked to see a plot of the mean age of breeding females through time and versus temperature, plots of breeding data v. temperature, etc - i.e. figures to represent the main pieces of the analytical results.

Thinking about the results from a life history perspective (and not knowing much about seals), I would have guessed that a) the decision to breed depends on energy stores, b) warmer winters are less energetically costly, and c) older, larger individuals are able to store more energy such that they have larger reserves in the spring and are able to breed earlier. This sort of pattern has been found in many species ranging from birds to fishes. In light of this, it would be terrific if the authors could provide a broader taxonomic context for their results. This would help expand the relevance of the temperature / age-structure / phenology connection the authors describe beyond the focal taxa.

Given the life history perspective, I would have liked to see more specific results on breeding time v. body size, temperature, and age that allowed for interactions between these variables - the GAMs are flexible, but additivity is a very strong assumption here. Also the authors included lagged effects (last year's breeding date, etc) which is great. But the authors need to carefully consider what these might mean. E.g. the introduction of delays could be used to represent autocorrelated environments, energetic carry-overs, other missing stage variables, or (unaccounted for) species interactions. Some discussion of these seems warranted.

Decision letter (RSPB-2021-0550.R0)

12-Apr-2021

Dear Dr Bull:

I am writing to inform you that your manuscript RSPB-2021-0550 entitled "Climate causes shifts in grey seal phenology through modifying age structure" has, in its current form, been rejected for publication in Proceedings B.

This action has been taken on the advice of referees, who have recommended that substantial revisions are necessary. With this in mind we would be happy to consider a resubmission, provided the comments of the referees are fully addressed. However please note that this is not a provisional acceptance.

Sincerely,
Dr Maurine Neiman
<mailto:proceedingsb@royalsociety.org>

Associate Editor
Comments to Author:

Two expert reviewers have now reviewed your study on how climate is affecting grey seal phenology. Both reviewers found merit in the approach, and clearly valued the importance of the topic. Reviewer 2 also found the manuscript to be well-written. However, both reviewers also expressed some significant level of dissatisfaction with the manuscript. Reviewer 1 thought considerably more in-depth analyses and discussion were needed, and further suggested the inclusion of more data from other UK populations. Reviewer 2 also suggested some additional analyses and a more detailed interpretation of the approach that moved beyond seals. I generally agree with the reviewers. The recommended changes/revisions would lead to a much stronger contribution on

Reviewer(s)' Comments to Author:

Referee: 1

Comments to the Author(s)

The authors try to understand the main drivers behind change in phenology of grey seals. An interesting aspect of the study is coupling the results with a population model to test one of the hypothesis.

However, the study would benefit from additional, broader analysis and further clarification. Grey seals in the UK waters are an interesting subject to study due to its different population dynamics around the UK, with some places still showing almost exponential increase. As also recommended by SCOS, grey seals should be more studied on a broader, geographical range, rather than small isolated populations. It has been shown by many studies that these animals travel a lot between regions, something which authors also mention. This study would, therefore, widely benefit from expanding the currently used dataset to other areas of the UK, where number of seals is much higher. Especially that 'Welsh' grey seals are difficult to study due to their cave breeding behaviour. There is a long term monitoring of individual and population-level breeding rate and phenology in many places around the UK.

It is also difficult to understand the main message of the paper. In some sections the authors focus mainly on changes of SST, which are not very pronounced during the study period (eg Fig S1), in other section on changes in age structure of the population and it is, therefore, not clear how the authors separated the effect of one from another on the phenology.

The study is missing more in depth description (especially for non-seal specialists) how changes in SST (or any environmental change) actually affect age – structure of a population. More information on a potential effect of climate change on fecundity, pup, and juvenile survival would be useful.

Too little effort (in my opinion) is put into explaining how SST or other factors may or may not affect phenology: is it via changes in food availability/distribution/type, is it direct effect on implantation rate, could overall (or local Fig S2) increase in grey seal population affect phenology? Could you maybe use your population model to explore it further? Which life history event of grey seals is most susceptible to changes in SST (e.g. implantation, birth, lactation, survival to age 1) and how it can be translated into population dynamics?

Finally, I find it a little bit concerning that the parameters for the population model are based on a very different seal population than the study population. Are there any data available for the grey seals anywhere around the UK instead of using data from north American seals?

Detailed comments:

Lines 90-92: explain how climate change may affect population age structure

Lines 324-327 These sentences seem to be incomplete. [...] we predicted that warmer SST would be associated with older females giving birth, on average (individual adult dataset). Does it mean that higher proportion of older females would give birth than during colder years? As predicted, we found average age (of ?) increased with increasing SST ($F = 11.8$, $p = 0.002$, Figure 4). Do younger females have lower fertility or lower weaning success or are just not pregnant at all? I know these are not easy questions to answer

Figure 4. I don't find it clear (based on the figure) that increase in SST results in older females giving birth earlier. Please clarify how do you distinguish between the effect of age and the effect of SST. The slope of the regression on panel a is mainly driven by larger sample size for young females and small sample size for the older females. Although I know that aging adults is difficult only based on visual observation, it is hard to interpret whether a female 1 year after 1st sighting is a female of for example age 4 which would be her first potential breeding season, or an older, more experienced female. No information what colours represent is provided. Your data seem to show quite a large range and variability, yet your CIs are very small.

Line 340. I can't see empirical population size on Figure S3, left panel

Figure 5. You mention three, I can only see two. Not clear what the scale (-0.012 to 0.012) represents. Usually sensitivity is represented as percentage change in relation to final settings of the model.

Line 403. Not really clear what [1] refer to. I guess your 1) in the above paragraph but please clarify

Figure S3. The ribbons are not visible. Are CI so small? It would be great to couple this graph with temporal changes in SST. The midpoint data and the length of the season show a cyclical trend. Could you explain it?

Figure 2: add that y axis show number of pups born per day

Figure 3. What do different colours of points represent? I guess range of temperature but please clarify

A circular graph showing timing of life events would be helpful to show when do pupping, implantation and weaning occur in relation to your quarterly SST. For an example see Fig. 1 in Silva et al (2020), *Environment International* 145

Supplementary material:

Study species section is not referred to in the main text

They show the probability of giving birth is 0.731 for females aged between 4-15 years and 0.831 for those aged between 16-26 years. In our model we assumed used the former figure (0.731) for the first two age classes of breeders, and we used the average of these for the oldest 'open' age group (0.781). How sensitive is your model to your assumptions of the lower age for the 'open' group?

Referee: 2

Comments to the Author(s)

In their manuscript "Climate causes shifts in grey seal phenology through modifying age structure" Bell and coauthors describe an analysis of long term monitoring data on pup production in grey seals in the Skomer Marine Conservation Zone, Wales. The authors find that the timing of pup production shifts with temperature, that older mothers breed earlier, and that warmer years were associated with older mothers. The authors constructed an age structured model to assess whether intrinsic changes in age structure could explain the shift and conclude that the observed changes must be driven by immigration / emigration.

Overall, this manuscript is well-written and most of the analyses seem reasonable. That said, the figures provided feel a bit incomplete. For instance, I would have liked to see a plot of the mean age of breeding females through time and versus temperature, plots of breeding data v. temperature, etc - i.e. figures to represent the main pieces of the analytical results.

Thinking about the results from a life history perspective (and not knowing much about seals), I would have guessed that a) the decision to breed depends on energy stores, b) warmer winters are less energetically costly, and c) older, larger individuals are able to store more energy such that they have larger reserves in the spring and are able to breed earlier. This sort of pattern has been found in many species ranging from birds to fishes. In light of this, it would be terrific if the authors could provide a broader taxonomic context for their results. This would help expand the relevance of the temperature / age-structure / phenology connection the authors describe beyond the focal taxa.

Given the life history perspective, I would have liked to see more specific results on breeding time v. body size, temperature, and age that allowed for interactions between these variables – the GAMs are flexible, but additivity is a very strong assumption here. Also the authors included lagged effects (last year's breeding date, etc) which is great. But the authors need to carefully consider what these might mean. E.g. the introduction of delays could be used to represent autocorrelated environments, energetic carry-overs, other missing stage variables, or (unaccounted for) species interactions. Some discussion of these seems warranted.

Author's Response to Decision Letter for (RSPB-2021-0550.R0)

See Appendix A.

RSPB-2021-2284.R0

Review form: Reviewer 1

Recommendation

Accept with minor revision (please list in comments)

Scientific importance: Is the manuscript an original and important contribution to its field?

Good

General interest: Is the paper of sufficient general interest?

Good

Quality of the paper: Is the overall quality of the paper suitable?

Good

Is the length of the paper justified?

Yes

Should the paper be seen by a specialist statistical reviewer?

No

Do you have any concerns about statistical analyses in this paper? If so, please specify them explicitly in your report.

No

It is a condition of publication that authors make their supporting data, code and materials available - either as supplementary material or hosted in an external repository. Please rate, if applicable, the supporting data on the following criteria.

Is it accessible?

Yes

Is it clear?

Yes

Is it adequate?

Yes

Do you have any ethical concerns with this paper?

No

Comments to the Author

Dear Authors,

You have done a really good job clarifying the manuscript. It is now much easier to understand why and how changes in SST may or may not affect pupping and why you chose the stated hypothesis regarding age structure.

I only have a minor comment to add. You write

'While we do not have data to test whether it is local demography or movement that changes age structure, we focus our modelling on whether stable age structure can account for our findings and discuss the likely role of movement in explaining that'.

Your manuscript would really benefit from adding one short paragraph to the discussion on suggestion on follow up studied which could dive deeper into understanding whether it is movement or local demography. How would you design a study and what kind of data would be required to answer this question?

Decision letter (RSPB-2021-2284.R0)

02-Nov-2021

Dear Dr Bull

I am pleased to inform you that your manuscript RSPB-2021-2284 entitled "Climate causes shifts in grey seal phenology through modifying age structure" has been accepted for publication, pending minor revisions, in Proceedings B.

Your manuscript has now been peer reviewed, and I assessed the review. The referee comments (not including confidential comments to the Editor) are included at the end of this email for your reference. As you will see, the referee was generally very positive about the revised manuscript, but nevertheless did make an additional fairly minor suggestion. I agreed with the referee, and I would like to invite you to revise your manuscript to make this change. Because the schedule for publication is very tight, it is a condition of publication that you submit the revised version of your manuscript within 7 days. If you do not think you will be able to meet this date please let us know.

- 1) A text file of the manuscript (doc, txt, rtf or tex), including the references, tables (including captions) and figure captions. Please remove any tracked changes from the text before submission. PDF files are not an accepted format for the "Main Document".
- 2) A separate electronic file of each figure (tiff, EPS or print-quality PDF preferred). The format should be produced directly from original creation package, or original software format. PowerPoint files are not accepted.
- 3) Electronic supplementary material: this should be contained in a separate file and where possible, all ESM should be combined into a single file. All supplementary materials

accompanying an accepted article will be treated as in their final form. They will be published alongside the paper on the journal website and posted on the online figshare repository. Files on figshare will be made available approximately one week before the accompanying article so that the supplementary material can be attributed a unique DOI.

Sincerely,

Dr Maurine Neiman

Reviewer(s)' Comments to Author:

Referee: 1

Comments to the Author(s).

Dear Authors,

You have done a really good job clarifying the manuscript. It is now much easier to understand why and how changes in SST may or may not affect pupping and why you chose the stated hypothesis regarding age structure.

I only have a minor comment to add. You write

'While we do not have data to test whether it is local demography or movement that changes age structure, we focus our modelling on whether stable age structure can account for our findings and discuss the likely role of movement in explaining that'.

Your manuscript would really benefit from adding one short paragraph to the discussion on suggestion on follow up studied which could dive deeper into understanding whether it is movement or local demography. How would you design a study and what kind of data would be required to answer this question?

Author's Response to Decision Letter for (RSPB-2021-2284.R0)

See Appendix B.

Decision letter (RSPB-2021-2284.R1)

08-Nov-2021

Dear Dr Bull

I am pleased to inform you that your manuscript entitled "Climate causes shifts in grey seal phenology by modifying age structure" has been accepted for publication in Proceedings B.

Data Accessibility section

Open Access

Paper charges

Sincerely,
Proceedings B
mailto:proceedingsb@royalsociety.org

Appendix A

12-Apr-2021

Dear Dr Bull:

I am writing to inform you that your manuscript RSPB-2021-0550 entitled "Climate causes shifts in grey seal phenology through modifying age structure" has, in its current form, been rejected for publication in Proceedings B.

This action has been taken on the advice of referees, who have recommended that substantial revisions are necessary. With this in mind we would be happy to consider a resubmission, provided the comments of the referees are fully addressed. However please note that this is not a provisional acceptance.

Sincerely,

Dr Maurine Neiman

Associate Editor

Comments to Author:

Two expert reviewers have now reviewed your study on how climate is affecting grey seal phenology. Both reviewers found merit in the approach, and clearly valued the importance of the topic. Reviewer 2 also found the manuscript to be well-written. However, both reviewers also expressed some significant level of dissatisfaction with the manuscript. Reviewer 1 thought considerably more in-depth analyses and discussion were needed, and further suggested the inclusion of more data from other UK populations. Reviewer 2 also suggested some additional analyses and a more detailed interpretation of the approach that moved beyond seals. I generally agree with the reviewers. The recommended changes/revisions would lead to a much stronger contribution on

Reviewer(s)' Comments to Author:

We are very grateful for the time and consideration put in by the reviewers and yourself. We are pleased with the overall positive assessment of our work and for the opportunity to resubmit. We have made some substantial revisions, as well as addressing the specific details flagged by the reviewers. Below, we set out point-by-point responses and hope that these, along with our resubmitted manuscript, make a convincing case for publication in Proceedings B.

Referee: 1

Comments to the Author(s)

The authors try to understand the main drivers behind change in phenology of grey seals. An interesting aspect of the study is coupling the results with a population model to test one of the hypothesis. However, the study would benefit from additional, broader analysis and further clarification. Grey seals in the UK waters are an interesting

subject to study due to its different population dynamics around the UK, with some places still showing almost exponential increase. As also recommended by SCOS, grey seals should be more studied on a broader, geographical range, rather than small isolated populations. It has been shown by many studies that these animals travel a lot between regions, something which authors also mention. This study would, therefore, widely benefit from expanding the currently used dataset to other areas of the UK, where number of seals is much higher. Especially that 'Welsh' grey seals are difficult to study due to their cave breeding behaviour. There is a long term monitoring of individual and population-level breeding rate and phenology in many places around the UK. It is also difficult to understand the main message of the paper. In some sections the authors focus mainly on changes of SST, which are not very pronounced during the study period (eg Fig S1), in other section on changes in age structure of the population and it is, therefore, not clear how the authors separated the effect of one from another on the phenology. The study is missing more in depth description (especially for non-seal specialists) how changes in SST (or any environmental change) actually affect age – structure of a population. More information on a potential effect of climate change on fecundity, pup, and juvenile survival would be useful. Too little effort (in my opinion) is put into explaining how SST or other factors may or may not affect phenology: is it via changes in food availability/distribution/type, is it direct effect on implantation rate, could overall (or local Fig S2) increase in grey seal population affect phenology? Could you maybe use your population model to explore it further? Which life history event of grey seals is most susceptible to changes in SST (e.g. implantation, birth, lactation, survival to age 1) and how it can be translated into population dynamics? Finally, I find it a little bit concerning that the parameters for the population model are based on a very different seal population than the study population. Are there any data available for the grey seals anywhere around the UK instead of using data from north American seals?

We are very grateful for the reviewer's considered comments and suggestions. We have pulled out the specific points from their text above and responded below, as well as providing point-by-point responses to Detailed Comments.

"An interesting aspect of the study is coupling the results with a population model to test one of the hypothesis"

We are pleased that the reviewer picks this out as a strength. Our modelling extends the findings of our empirical study to help confer generality. A key aim with our study was to use this case study of Skomer seals to demonstrate a novel mechanism occurring in nature, but then make our findings more relevant and interesting to the broad readership of Proceedings B through this simple but widely applicable matrix population model. While the focal species of our study is the grey seal, our paper points to more broadly generalisable 'rules of nature' through population modelling.

"Especially that 'Welsh' grey seals are difficult to study due to their cave breeding behaviour".

It is true that around half of grey seal pups in North Wales are found in 'cryptic environments' that include caves (co-author Stringell et al., 2014). However, only a very small proportion of South Wales seals produce pups in caves. In fact, South Wales (particularly the Skomer Marine Conservation Zone) accounts for the substantial majority of grey seal pups in Wales. Around Skomer / Marloes, the number of pups in caves is a very small minority of animals.

"There is a long term monitoring of individual and population-level breeding rate and phenology in many places around the UK."

That is true; however, we have searched the literature and reached out to other seal researchers around the UK (one of our co-authors represents Wales on SCOS) and have not found anywhere with the intensity and length of monitoring undertaken at Skomer, so suitable to test our hypotheses. Our data have involved wardens recording animals every few days, for the entire pupping season, every year over nearly three decades. In parallel, they also mark and follow the fate of the adult female population at the same site for the same length of time, so that pup census data can be closely integrated with the individual adult longitudinal study. All the components of this type of monitoring do happen elsewhere, but we have not found anything quite comparable to our dataset in the UK.

More constructively, it is already known that there are phenological shifts, sometimes associated with aspects of climate, in grey seals (cited in our manuscript). Therefore, we feel that replicating our findings elsewhere amongst UK seals (if this were possible) is not as useful to a wider biological readership as exploring the generality of the mechanism using matrix population modelling (MPM). This was a key reason we developed our MPM, and some other reviewer suggestions have helped us improve that aspect of our study. To better support this aim, we have now included a wider range of demographic parameter estimates

for grey seals, from around the UK and further afield, both empirical and through model sensitivity analysis (new analysis from line 394 and new Figure 5b).

“In some sections the authors focus mainly on changes of SST, which are not very pronounced during the study period (eg Fig S1), in other section on changes in age structure of the population and it is, therefore, not clear how the authors separated the effect of one from another on the phenology.”

The fluctuations are much clearer, and mirror the phenological shifts, when viewed as just winter seasonal means. In response to another suggestion, these are now included in Figure 2. We are not quite sure what the reviewer intends by asking how we separate SST from age structure. The aim of our study is to test whether SST acts on phenology through modifying age structure. Hopefully some of the changes we've made in our revised manuscript, in response to other suggestions, make this clearer (e.g., new paragraph from line 102).

“The study is missing more in depth description (especially for non-seal specialists) how changes in SST (or any environmental change) actually affect age – structure of a population.

We now describe some of the key ways how age structure may be changed through local demographic processes, such as fecundity or differential survival across age classes, or through movement in and out of an open population (new text from line 102 (seals) and new text from line 516 (other species)).

More information on a potential effect of climate change on fecundity, pup, and juvenile survival would be useful.

Changes to fecundity or pup and juvenile survival are not really the focus of our study. Our main aim was to compare between the traditional theory of individual level effects (delayed implantation) and our new hypothesis about population level effects (average age of adult females). Since our empirical findings favour the adult age explanation for changes in pupping phenology, our modelling focus was then on adult survival. In fact, since our modelling sensitivity analysis suggests that local demographic processes cannot account for the observed phenological shift, we strongly suspect that movement patterns are the key to understanding the age structuring.

Too little effort (in my opinion) is put into explaining how SST or other factors may or may not affect phenology: is it via changes in food availability/distribution/type, is it direct effect on implantation rate, could overall (or local Fig S2) increase in grey seal population affect phenology?”

It would be a huge amount of work to relate food availability patterns to seal distributions and dynamics for Skomer seals, as it would be to investigate implantation rate etc. While interesting to do, it is beyond the scope of our current study and our key findings are not dependent on this. However, we have included some new discussion (from line 516), which covers this sort of explanation across a range of taxa.

On the reviewer's last point, it does seem that phenology shifts are not dependent on increases in grey seal population abundance (we included abundance as an explanatory variable, partly to control for this phenomenon).

“Which life history event of grey seals is most susceptible to changes in SST (e.g. implantation, birth, lactation, survival to age 1) and how it can be translated into population dynamics?”

This is beyond the scope of our study, and we think that it would lose focus. Hopefully this is covered by our responses above.

“I find it a little bit concerning that the parameters for the population model are based on a very different seal population than the study population. Are there any data available for the grey seals anywhere around the UK instead of using data from north American seals?”

We now cite Lonergan et al. (2011) and Thomas et al. (2019), both of which estimate grey seal demographic parameters around the UK. In fact, all three sources (with Bowen et al., 2006, for North America) show remarkable consistency in the parameters most relevant to our study, e.g., adult survival of around 0.9.

Detailed comments:

Lines 90-92: explain how climate change may affect population age structure

Please see responses above.

Lines 324-327 These sentences seem to be incomplete.

[...] we predicted that warmer SST would be associated with older females giving birth, on average (individual adult dataset). Does it mean that higher proportion of older females would give birth than during colder years?

We have changed “we predicted that warmer SST would be associated with older females giving birth, on average (individual adult dataset)” to “we predicted that warmer SST would be associated with an increase in the average age of females giving birth (individual adult dataset)” (new line 367).

As predicted, we found average age (of ?) increased with increasing SST ($F = 11.8$, $p = 0.002$, Figure 4). Do younger females have lower fertility or lower weaning success or are just not pregnant at all?

Average age “of adult females” now clarified on line 369. Fertility is hard to assess so we can't present data or analysis on that here. Weaning success was not quantified in our study (being not strictly relevant to a study on pupping phenology, although interesting for a follow-up study), but it is known to improve with experience generally in grey seals, as many other long-lived animals. Our analysis of adult females aimed to relate adult ageing (since first seen) to pupping date, so these are necessarily pregnant females. It is not known whether additionally the pregnancy rate of Skomer seals changes with age.

I know these are not easy questions to answer

True! While they are all interesting in their own right, we think that the main message of our study – that age structure, rather than individual response, is a better explanation of phenology shifts in this species – is still an important contribution to our understanding of how long-lived species may be impacted by climate change.

Figure 4. I don't find it clear (based on the figure) that increase in SST results in older females giving birth earlier. Please clarify how do you distinguish between the effect of age and the effect of SST. The slope of the regression on panel a is mainly driven by larger sample size for young females and small sample size for the older females. Although I know that aging adults is difficult only based on visual observation, it is hard to interpret whether a female 1 year after 1st sighting is a female of for example age 4 which would be her first potential breeding season, or an older, more experienced female.

This is an important error on our part and we are so grateful that the reviewer spotted it! At some point in the final assembly of the figures, the critical Figure 4c (mother age vs. SST) got replaced with another (non-significant) relationship. The relationship is indeed positive and significant (as already indicated in our statistical text), so that warmer SST is clearly associated with older adult females. We have amended Figure 4c in our revised manuscript. Hopefully the entire thrust of our argument is now much clearer and stronger.

The second point the reviewer makes about the relationship in Figure 4a being based on unequal sample sizes is not the case. The 95% confidence ribbon here takes this sampling coverage into account (widening with increasing age) but still shows a robust, negative relationship.

The final point about first sighting not necessarily equating to a particular age is also a very good one. However, we dealt with this statistically by modelling individuals with random intercepts in our mixed effects model (Methods: Statistical analysis, line 249). So, for a given adult, we can see that ‘ageing’ (not ‘age’) is associated with earlier pupping, regardless of actual age (the height on the plot; intercept).

No information what colours represent is provided.

We have now explained that the colours in Figure 4b and 4c relate to SST. We have removed the colour gradient (changing with adult female age) on Figure 4a as this was more decorative than informative.

Your data seem to show quite a large range and variability, yet your CIs are very small.

The CIs are marginal, rather than conditional, so indicate confidence in the population level outcome (i.e., partitioning out the differences between individuals as a random effect). This explains why they appear relatively small compared to the spread of the data. We regard this as the more appropriate illustration of certainty as we are asking a population level question rather than focussing on particular individuals. We have now added the word 'marginal' where appropriate for clarity (Figures 3 and 4).

Line 340. I can't see empirical population size on Figure S3, left panel

We're not quite sure what empirical population size is missing here. The dependent variable is the pup population size. This looks clear to us but we're happy to provide further clarification if requested. To help identify the figure in question, original Figure S3 is now Figure 2 in the main text and has been modified, which may in itself deal with this point.

Figure 5. You mention three, I can only see two.

“three” amended to “two”.

Not clear what the scale (-0.012 to 0.012) represents. Usually sensitivity is represented as percentage change in relation to final settings of the model.

This was shown as proportional change but is now amended to percentage change in our revised Figure 5.

Line 403. Not really clear what [1] refer to. I guess your 1) in the above paragraph but please clarify

Correct. Now clarified on lines 458-470.

Figure S3. The ribbons are not visible. Are CI so small?

The confidence interval ribbons were missing on Figure S3. We included them in our submitted figure, but it seems that they disappeared when the final pdf was generated as part of the online submission process. We missed this and apologise for the oversight. This is now Figure 2 in the main document. We will check this carefully when we submit our revised manuscript. We just have to hope they reach reviewers with grey confidence interval ribbons clearly visible on all panels.

It would be great to couple this graph with temporal changes in SST.

This is an excellent suggestion. The new version includes winter SST and you can see the effect of temperature on season midpoint (panel b) in particular. As a result of this helpful insight about how to present our findings, we have now promoted this to Figure 2 in the main Results, rather than Supplementary.

The midpoint data and the length of the season show a cyclical trend. Could you explain it?

It's certainly tempting to see a cycle here but at just over one putative cycle long, we think it is premature to interpret it that way. You can now see that midpoint mirrors the long-term fluctuations in SST (new Figure 2 at the reviewer's suggestion), which further demonstrates our finding that warmer years are later years.

Figure 2: add that y axis show number of pups born per day

This is now clarified in the legend.

Figure 3. What do different colours of points represent? I guess range of temperature but please clarify

Correct. Now clarified in the legend.

A circular graph showing timing of life events would be helpful to show when do pupping, implantation and weaning occur in relation to your quarterly SST. For an example see Fig. 1 in Silva et al (2020), Environment International 145

We agree this is a useful addition. For efficiency of publication space, we have included a circular graph relating quarterly SST seasons to pupping on our map, Figure 1. We felt that further breakdown into implantation and weaning would not be helpful, as these overlap substantially with pupping (e.g., weaning approximately three weeks after birth, over a 4-month pupping season). We hope this revision now gives a quick, visual guide to timing, particularly for a broad biological readership not familiar with this species.

Supplementary material: Study species section is not referred to in the main text

Now referred to at the end of Methods: Seal data.

They show the probability of giving birth is 0.731 for females aged between 4-15 years and 0.831 for those aged between 16-26 years. In our model we assumed used the former figure (0.731) for the first two age classes of breeders, and we used the average of these for the oldest 'open' age group (0.781). How sensitive is your model to your assumptions of the lower age for the 'open' group?

In fact, our model is very insensitive to this assumption. As can be seen from our Figure 5, increasing birth rates for each age class in isolation by 10% results in a later pupping season. Therefore, lowering the age of the 'open' group, so that the 5-6 transition has the same, higher, birth rate as the 6+ group, would lead to a later pupping season but only very slightly (less than 0.1 hours). We also tried decreasing each birth rate by 10% and found the expected advance, but to a maximum of 1% (compared with 7.5% shift due to changing oldest adult survival by 10%). However, we did not include this analysis here, as our empirical findings point to the hypothesis that the age distribution of breeding adults is the mechanism, not the relative fecundity of those adults. Specifically, we wanted to test whether changes to the stable age distribution could account for our observed phenological changes, or whether transient changes to age distribution (e.g., through immigration) would be required. It turns out the latter is true. We have now made this aim more explicit to avoid confusion (line 125).

Also, we have removed the redundant word "used" in the cited text (was "assumed used").

Referee: 2

Comments to the Author(s)

In their manuscript "Climate causes shifts in grey seal phenology through modifying age structure" Bell and coauthors describe an analysis of long term monitoring data on pup production in grey seals in the Skomer Marine Conservation Zone, Wales. The authors find that the timing of pup production shifts with temperature, that older mothers breed earlier, and that warmer years were associated with older mothers. The authors constructed an age structured model to assess whether intrinsic changes in age structure could explain the shift and conclude that the observed changes must be driven by immigration / emigration.

Overall, this manuscript is well-written and most of the analyses seem reasonable.

We are grateful for the reviewer's positive and accurate summary of our work. Below, we provide point-by-point responses to their specific suggestions.

That said, the figures provided feel a bit incomplete. For instance, I would have liked to see a plot of the mean age of breeding females through time and versus temperature, plots of breeding data v. temperature, etc - i.e. figures to represent the main pieces of the analytical results.

If we understand the reviewer's suggestions correctly, actually some of that is already included. Mean age of females vs. temperature is our Figure 4c. Plots of "breeding data" vs. temperature are shown for adult

females in our Figure 4b and for pups in Figure 3. We have now signposted the relevant figures more clearly in the text (lines 356, 358, 370). The reviewer is correct that these relationships are the main pieces of our empirical analysis.

The suggestion of investigating adult female age through time is a good one. We have now included the findings of that analysis in our revised manuscript (line 350). However, the relationship was not statistically significant, and this analysis is more of a control against biases through time than a core part of our hypothesised mechanism about age structure. Therefore, we have not included a figure, in the interests of clarity and space.

Thinking about the results from a life history perspective (and not knowing much about seals), I would have guessed that a) the decision to breed depends on energy stores, b) warmer winters are less energetically costly, and c) older, larger individuals are able to store more energy such that they have larger reserves in the spring and are able to breed earlier. This sort of pattern has been found in many species ranging from birds to fishes. In light of this, it would be terrific if the authors could provide a broader taxonomic context for their results. This would help expand the relevance of the temperature / age-structure / phenology connection the authors describe beyond the focal taxa.

Thanks for the useful suggestion. We have now included a new penultimate paragraph specifically discussing the relationship between age structure, climate change, and phenology in birds, fish, and mammals. In fact, as the reviewer notes by citing birds and fishes as examples, there is much less published work on this in mammals, making the advance made by our study an important one, in our view.

Given the life history perspective, I would have liked to see more specific results on breeding time v. body size, temperature, and age that allowed for interactions between these variables – the GAMs are flexible, but additivity is a very strong assumption here. Also the authors included lagged effects (last year's breeding date, etc) which is great. But the authors need to carefully consider what these might mean. E.g. the introduction of delays could be used to represent autocorrelated environments, energetic carry-overs, other missing stage variables, or (unaccounted for) species interactions. Some discussion of these seems warranted.

We don't have data on body size but since the relationship between pupping date and adult age (Figure 4a) and between pupping date and SST (Figure 4b) are both linear on a linear predictor scale (tested using a GAM but the resulting best fit was linear), we are able to test the statistical interaction between SST and age (so the idea that older adults may be less sensitive to SST than younger ones). In the event, the interaction is not statistically significant ($\Delta AICc = 1.93$). We now report this finding in our revised manuscript (line 354).

The reviewer is also correct to point out the range of possible causes explaining the lagged effect. Primarily, this was included to statistically control of any or all of these, rather than being a focus of our hypotheses. However, the reviewer is also correct that there are numerous different possible causes of lagged effects. In fact, all those suggestions are plausible, and this wasn't a key focus of our study. We prefer to simply note these potential causes in our Methods (line 214) in our revised manuscript, rather than add slightly off-topic speculation in our Discussion.

Appendix B

6th November 2021

Dear Dr Neiman,

We are delighted to read that our paper has been accepted, pending one minor revision: *“Your manuscript would really benefit from adding one short paragraph to the discussion on suggestion on follow up studied which could dive deeper into understanding whether it is movement or local demography. How would you design a study and what kind of data would be required to answer this question?”*

We have now addressed this by inclusion of a new Discussion paragraph. We append a revised version of our manuscript here, with this as a tracked change.

Many thanks,

Jim

James Bull, PhD SFHEA | Associate Professor | Biosciences | Swansea University